# Context-Aware Emotion Recognition via Multi-View Instruction-Tuned Visual Language Guidance

## Abstract

Context-aware emotion recognition often relies on heterogeneous cues, but many state-of-the-art systems still hinge on engineered signals (e.g., pose landmarks or temporal cues), limiting applicability. Meanwhile, VLM-based emotion recognition remains relatively under-explored in current research. Our work targets this gap with a parameter-efficient, interpretable design. To mitigate class imbalance and make view–emotion relations explicit, we first curate an LLM-assisted QA dataset. In Stage 1, the VLM is adapted into a multi-view emotion encoder that extracts fine-grained features from scene, body, and face using shared, parameter-efficient components with view-specific pathways, enabling interpretable evidence disentanglement from a single image. In Stage 2, the VLM remains frozen and its scene/body/face descriptors are fused by a lightweight head. This preserves VLM knowledge (avoiding overfitting and label coupling) while yielding independent, well-calibrated scores that support flexible thresholds, plug-and-play label sets, and strong sample efficiency. Using only single-image inputs, our pipeline attains 37.88 mAP on EMOTIC, 88.82% top-1 accuracy on CAER-S, and higher recall/F1 on HECO than prior VLM-based baselines, while offering clear per-view interpretability. Code, prompts, and data splits will be released.

## 1 Introduction

Understanding human emotions through visual cues is a central challenge in computer vision with significant implications for human-computer interaction, robotics, and intelligent surveillance (e.g., Cowie et al., 2001; Ziemke, 2008). While early research predominantly focused on isolated facial expressions (e.g., Ntizikira et al., 2025)., it is now widely recognized that humans interpret emotions holistically, integrating crucial contextual signals from body language, gestures, and the surrounding environment (e.g., Jiang et al., 2020; Dhall et al., 2012).

Many prior works have attempted to integrate this visual context, from leveraging multi-modal features (e.g., Barrett et al., 2011) to employing graph-based methods for modeling object relations (e.g., Liu et al., 2023a). Despite these advances, recognizing complex or ambiguous emotions remains a formidable challenge. The advent of Vision-Language Models (VLMs) offers a promising direction, leveraging their powerful semantic reasoning capabilities (e.g., Kosti et al., 2017). However, directly fine-tuning these generative models for emotion recognition presents significant hurdles: they often lack the structured, probabilistic outputs required by downstream systems and suffer from prompt sensitivity, which hinders their reliable deployment. This is because autoregressive models (such as VLMs or LLMs) generate labels sequentially, conditioning each step on the previous outputs, which is problematic for tasks that require independent per-label confidence scores.

To bridge this critical gap, we propose a two-stage framework that synergistically combines the rich semantic understanding of VLMs with the robust, controllable nature of dedicated classifiers. In the first stage, we instruction-tune a pre-trained VLM to specialize in emotion-centric reasoning, guiding it to focus on the interplay between facial expressions, body language, and the scene. This process transforms the generic VLM into an expert feature

extractor for emotion. In the second stage, to overcome the limitations of generative outputs, we freeze the VLM, extract its now-specialized visual-language representations, and train a lightweight classifier that yields calibrated, probabilistic scores for each emotion category.

In summary, our main contributions are:

- We propose a two-stage framework that repurposes a generative VLM into a reliable feature extractor, harnessing its semantic power while ensuring controllable, probabilistic outputs suitable for real-world applications.
- We demonstrate that targeted instruction tuning can effectively align a VLM's representations with nuanced emotional semantics, enhancing its ability to decipher complex contextual cues.
- Our method achieves competitive performance using only three foundational visual inputs, demonstrating that sophisticated feature engineering can be supplanted by effective VLM adaptation.

## 2   Related Work

### 2.1   Context-Aware Emotion Recognition

Traditional emotion recognition primarily focuses on facial expressions, while recent studies emphasize contextual cues such as background, human-object interaction (HOI), and human-human interaction (HHI). Early works combined body and scene features Kosti et al. (2017); Mittal et al. (2020); Lee et al. (2019); Li et al. (2021), while later studies introduced object-level context modeling with attention mechanisms Yang et al. (2022); Jiang et al. (2020); Li et al. (2021) and inter-agent reasoning Mittal et al. (2020); Hoang et al. (2021). These efforts significantly improved recognition accuracy, though often at the cost of higher model complexity and reduced practicality.

### 2.2   Vision-Language Models

VLMs jointly learn from images and text, with early works such as CLIP Radford et al. (2021) and BLIP Li et al. (2022) establishing strong multimodal representations. Recent models (InstructBLIP, LLaVA, Qwen-VL, GPT-4V, Gemini, InternVL) further integrate VLMs with LLMs, leveraging projectors or Q-Former modules for multimodal reasoning and instruction following. In emotion recognition, EmotionCLIP directly maps images to emotion-related text Zhang et al. (2023), while other approaches use captioning pipelines or few-shot CoT reasoning with large VLMs Lei et al. (2024). These works highlight VLMs as promising tools for context-aware emotion recognition.

### 2.3   Instruction Tuning for Multimodal Models

Instruction tuning enhances model generalization by aligning outputs with natural language prompts. In NLP, FLAN Wei et al. (2021) demonstrated strong zero-shot performance across tasks. Extending to multimodal settings, InstructBLIP Dai et al. (2023) applied instruction tuning while freezing the vision encoder and LLM, outperforming BLIP-2 Li et al. (2023a) and Flamingo Alayrac et al. (2022). These results show instruction tuning as a key enabler of multi-task VLMs.

### 2.4   Handling Data Imbalance

Emotion recognition datasets are often imbalanced across classes. Sampling strategies (e.g., SMOTE, data augmentation, undersampling) help rebalance training sets, while loss designs such as class-balanced loss Cui et al. (2019), focal loss Lin et al. (2017), and asymmetric loss Ridnik et al. (2021) emphasize minority or hard-to-classify samples. Combining these techniques with multimodal models remains a challenge for robust real-world deployment.

The advent of Vision-Language Models (VLMs) has opened a new paradigm for CAER. Foundational models like CLIP (Radford et al., 2021) and BLIP (Li et al., 2022) achieved

robust cross-modal semantic alignment. More recently, advanced models such as Instruct-BLIP (Dai et al., 2023) and LLaVA (Liu et al., 2023b) have demonstrated remarkable visual reasoning and instruction-following capabilities by coupling vision encoders with Large Language Models (LLMs) (Chiang et al., 2023). Researchers have begun applying VLMs to emotion recognition, for instance, by leveraging zero-shot or few-shot prompting with large models like GPT-4V (Etesam et al., 2024). Although promising, these approaches highlight a critical limitation for practical deployment: they typically lack structured, probabilistic outputs and exhibit significant sensitivity to minor changes in prompts (Etesam et al., 2024).

As summarized in Table 1, prior CAER approaches tend to combine increasingly complex sets of features, from scene and face to HOI, HHI, or depth cues. In contrast, our method shows that a lightweight feature combination—limited to scene, face, and body cues—when guided by instruction-tuned VLMs, can already achieve state-of-the-art performance. This observation motivates our two-stage framework described in the Section 3.

Table 1: Comparison of context-aware emotion recognition methods based on feature types used

| Method | Scene / BG | Face | Body | Pose / Gait | HOI | HHI | Depth Map |
|---|---|---|---|---|---|---|---|
| Kosti et al. Kosti et al. (2017) | ✓ | | ✓ | | | | |
| Lee et al. Lee et al. (2019) | ✓ | ✓ | | | | | |
| Zhang et al. Zhang et al. (2019) | ✓ | | | | ✓ | | |
| Gao et al. Gao et al. (2021) | ✓ | | | | ✓ | | |
| Mittal et al. Mittal et al. (2020) | ✓ | ✓ | | ✓ | | | ✓ |
| Li et al. Li et al. (2021) | ✓ | ✓ | ✓ | | | | |
| Hoang et al. Hoang et al. (2021) | ✓ | ✓ | ✓ | | ✓ | | |
| Li et al. Li et al. (2023b) | ✓ | | ✓ | | ✓ | | |
| Mittal et al. Mittal et al. (2021) | ✓ | ✓ | | ✓ | | | ✓ |
| Mittel et al. Mittel & Tripathi (2023) | ✓ | ✓ | | | | | |
| Yang et al. Yang et al. (2022) | ✓ | ✓ | | ✓ | ✓ | ✓ | |
| Ours | ✓ | ✓ | ✓ | | | | |

Note: HOI = Human-Object Interaction; HHI = Human-Human Interaction. Ours leverages only scene, face, and body features, yet achieves state-of-the-art performance through instruction-tuned VLM guidance.

## 3 Method

### 3.1 Datasets, Balancing, and Instruction Tuning (Brief)

We use EMOTIC (26 multi-label), CAER-S (7 balanced), and HECO (8 moderately imbalanced). For EMOTIC/HECO we apply label-aware balancing (minor-label QA augmentation; major-label downsampling), while CAER-S is unchanged. We also build a lightweight instruction-tuning corpus (caption-guided classification, visual description, and rationale QA) to unify supervision. Full details and distributions are in Appendix A.

### 3.2 Overview of Two-Stage Architecture

Our architecture consists of two stages. In the first stage, we extend InstructBLIP into a multi-view VLM by integrating three complementary types of visual semantic extractors: scene-aware, body-aware, and face-aware. Each extractor focuses on a specific aspect of emotional cues, including scene context, body posture, and facial expression. These components are instruction-tuned to align with emotion-related tasks and generate meaningful representations. In the second stage, the trained VLM serves as a frozen emotion-aware feature extractor. We concatenate features from all three extractors and pass them through a classifier. This classifier outputs a confidence score for each emotion label, supporting multi-label prediction with interpretability and downstream compatibility.

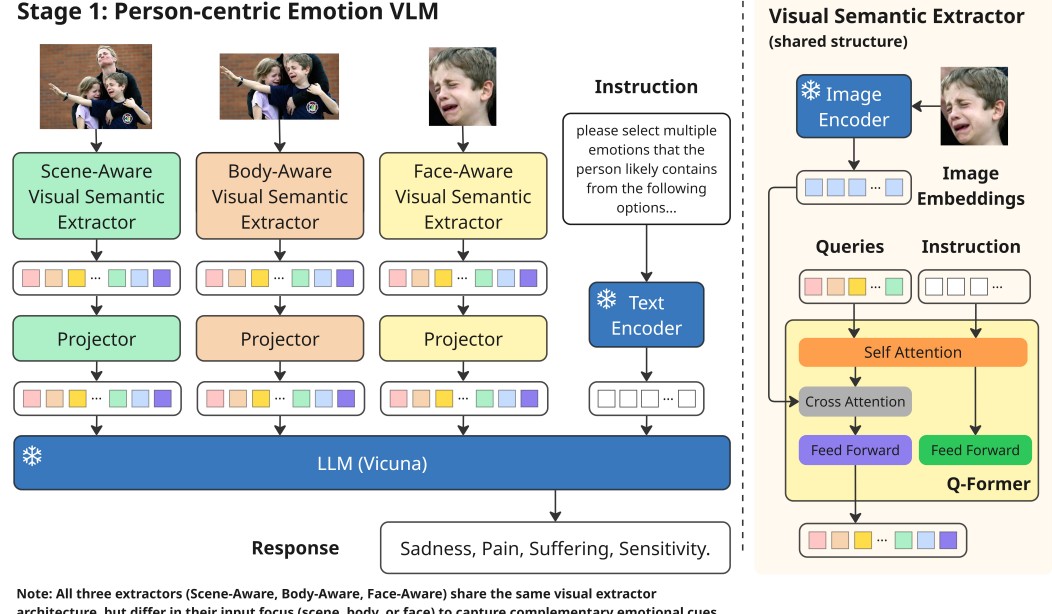

Figure 1: Overview of Stage 1 – Person-centric Emotion VLM. The left illustrates the overall architecture, which includes three types of visual semantic extractors—scene-aware, body-aware, and face-aware—designed to capture emotional cues. The right shows the shared structure of a visual semantic extractor.

## 3.3 Stage 1: Person-centric Emotion VLM

We extend InstructBLIP Dai et al. (2023) to extract emotion-relevant features from three visual perspectives. The scene-aware module captures contextual cues from the background and environment. The body-aware module focuses on posture and gesture information. The face-aware module encodes facial expressions and local cues. Each module uses a frozen image encoder (ViT-g/14), a shared Q-Former with distinct query tokens, and a projection layer.

The Q-Former inherits pre-trained weights from InstructBLIP. During fine-tuning, we update the Q-Former, the module-specific query tokens, and the projection layers, enabling the model to align with emotion-centric visual information. To support the face and body-aware visual semantic extractors, we require bounding boxes for the subject's body and face. If these annotations are unavailable in the dataset, we use YOLOv11 Khanam & Hussain (2024) to estimate them.

We apply instruction tuning to the VLM to improve its understanding of emotion-related context. This tuning involves using task-specific prompts and supervision that guide the model to better recognize and describe emotions across different visual perspectives. More details about the instruction tuning dataset are provided in Section A.3.

After Stage 1 training, the VLM can perform visual question answering (VQA) for emotion-related prompts and serves as a strong feature extractor for the classification task in Stage 2.

## 3.4 Model Architecture

Scene/Body/Face-Aware Visual Semantic Extractor Our model employs three visual semantic extractors—scene-aware, body-aware, and face-aware—to capture different emotional cues. All extractors share the same backbone architecture, consisting of a frozen

image encoder, a Q-Former, and a projection layer. Each extractor differs in its input region, query tokens, and projection head.

The scene-aware module processes the entire image to capture contextual information such as environment and lighting. The body-aware and face-aware modules operate on cropped regions, obtained using YOLOv11 Khanam & Hussain (2024) when annotations are unavailable. The body module captures posture and gestures, while the face module focuses on fine-grained expressions.

To reduce training cost, the body-aware and face-aware modules share the Q-Former weights but use distinct query tokens and projection heads to preserve modality-specific expressiveness.

**Image Encoder**  We adopt the ViT-g/14 transformer Fang et al. (2023) as the backbone for visual encoding. It is kept frozen throughout both stages to preserve general-purpose image representations. Each extractor uses the encoder to obtain feature embeddings, ensuring consistency across modules.

**Large Language Model**  For instruction-tuning and downstream reasoning tasks, we use Vicuna-7B Chiang et al. (2023), a decoder-only transformer derived from LLaMA Touvron et al. (2023). It remains frozen during training, allowing us to focus on adapting the vision side without overfitting the language model.

## 3.5  Stage 2: Method

**Stage 2: Emotion Classifier**

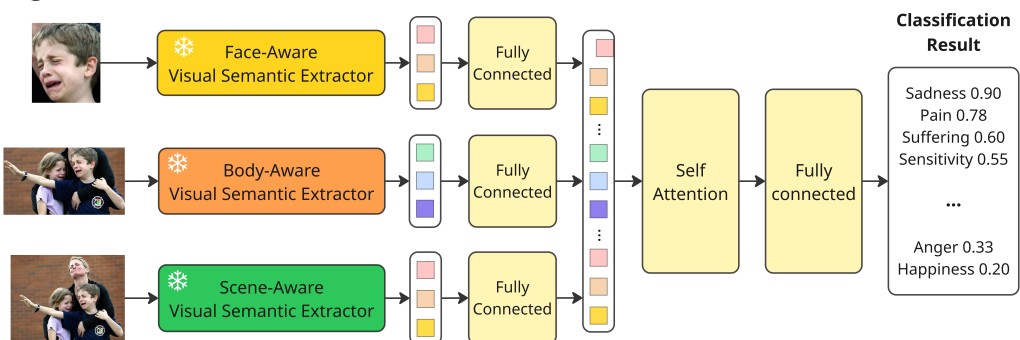

Figure 2: Overview of Stage 2. The VLM-extracted visual features are projected, fused via self-attention, and used for emotion classification.

In the second stage, we utilize the pretrained vision-language model (VLM) as a fixed feature extractor to obtain high-level semantic features from three different perspectives: face-aware, body-aware, and scene-aware representations. Each of these visual features captures distinct but complementary information related to human emotion.

To integrate these representations into a shared semantic space, each modality-specific feature is passed through its own fully connected layer. This projection step not only aligns the representations but also reduces their dimensionality into a more compact form suitable for interaction.

After projection, the three modality representations are stacked together to form a short sequence. This sequence is then processed by a self-attention module, which allows the model to capture inter-modality relationships by dynamically attending to the most informative features across face, body, and scene.

Finally, the output from the self-attention layer is then flattened into a single unified vector and fed into another fully connected layer that maps it to a fixed-size emotion prediction

vector. This output contains confidence scores for each emotion label, supporting emotion classification via threshold-based decision making.

## 3.6 Loss Function

Our training procedure involves two distinct objectives across the two stages of the proposed architecture: instruction-tuning and classification.

### 3.6.1 Stage 1: Instruction-Tuning Objective

In the first stage, we fine-tune the vision-language model to follow emotion-related prompts through an instruction-tuning objective. Given a ground-truth token sequence $y = (y_1, \ldots, y_T)$ and the model-predicted probability distribution $p = (p_1, \ldots, p_T)$ over the vocabulary at each time step, we apply the standard cross-entropy loss:

$$\mathcal{L}_{\mathrm{CE}} = -\sum_{t=1}^{T} \log p_t(y_t). \tag{1}$$

This loss encourages the model to generate accurate textual responses aligned with the provided instructions.

### 3.6.2 Stage 2: Classification Objective

In the second stage, we train a classifier using the extracted features. The specific loss function depends on the label type of the downstream dataset.

**Multi-Label Classification: Asymmetric Loss** For datasets with multi-label emotion annotations, where each sample can be associated with multiple emotion classes, we adopt the asymmetric loss (Ridnik et al., 2021), which is designed to handle label imbalance effectively. Given a predicted probability $p \in [0,1]$ for a single class and the corresponding ground-truth label $y \in \{0,1\}$, the loss for each instance is defined as:

$$\mathrm{ASL}(p,y) = \begin{cases} L_+ = (1-p)^{\gamma_+} \log(p), & \text{if } y = 1, \\ L_- = (p_m)^{\gamma_-} \log(1-p_m), & \text{if } y = 0, \end{cases} \tag{2}$$

$$p_m = \max(p - m, 0), \tag{3}$$

where $\gamma_+$ and $\gamma_-$ are focusing parameters for positive and negative samples, respectively, and $m \in [0,1]$ is a probability margin that controls the aggressiveness of filtering easy negatives. The total loss over $C$ labels is

$$\mathcal{L}_{\mathrm{asym}} = -\sum_{i=1}^{C} \mathrm{ASL}(p_i, y_i). \tag{4}$$

**Multi-Class Classification: Cross-Entropy Loss** For multi-class emotion datasets, we apply the standard categorical cross-entropy loss. Let $y_i \in \{0,1\}$ be the one-hot encoded ground-truth label and $p_i$ the predicted probability for class $i$ with $\sum_i y_i = 1$:

$$\mathcal{L}_{\mathrm{CE}} = -\sum_{i=1}^{C} y_i \log(p_i). \tag{5}$$

This formulation encourages the classifier to assign high probability to the correct single class for each input sample.

## 4 Experiments

### 4.1 Experiment Settings

We train our model in two stages. Stage-1 (instruction tuning): we fine-tune only the Q-Former modules in the visual semantic extractor on our QA-style emotion understanding corpus, while freezing the image encoder and the LLM. Stage-2 (classifier): we freeze

the vision-language backbone and train a lightweight classifier; model selection is based on validation mAP for multi-label datasets and accuracy for multi-class datasets. Unless otherwise specified, results are reported with the best validation checkpoint. Additional implementation details and hyperparameters are provided in the Appendix.

## 4.2 Evaluation Metrics

We evaluate model performance using standard metrics for both multi-class and multi-label emotion recognition. For multi-class datasets (e.g., CAER-S), we report overall classification accuracy. For multi-label datasets (e.g., EMOTIC, HECO), we report macro-averaged precision, recall, F1-score, Hamming Loss, and mean Average Precision (mAP). Among these, mAP serves as our primary evaluation metric, as it captures both precision and recall across confidence thresholds.

## 4.3 Main Results

We evaluate our method on three benchmarks: EMOTIC, CAER-S, and HECO. Table 2 shows results on EMOTIC, where our classifier achieves the best mAP (37.88), surpassing prior methods including Yang et al. Yang et al. (2022), which use more feature types.

Table 2: Comparison of Performance on the EMOTIC Dataset.

| Method | mAP ↑ | Precision ↑ | Recall ↑ | F1 Score ↑ |
|---|---|---|---|---|
| Classifier-Based Methods | | | | |
| Kosti et al. (Kosti et al., 2017) | 28.33 | 25.02 | 35.07 | 28.83 |
| Mittal et al. (Mittal et al., 2020) | 35.48 | - | - | - |
| Yang et al. (Yang et al., 2022) | 37.73 | - | - | - |
| Ours (Classifier) | 37.88 | 37.50 | 44.04 | 38.35 |
| VLM/LLM-Based Methods | | | | |
| Etesam et al. (Etesam et al., 2024) (GPT-4V) | - | 37.48 | 38.35 | 34.47 |
| Ours (VLM)* (7B) | - | 33.58 | 40.78 | 33.65 |

* indicates fine-tuned models.

In the VLM setting, our 7B model achieves the highest recall (40.78) and competitive F1 compared to GPT-4 Vision, despite being significantly smaller. Category-wise results 3 show SOTA performance on 8 out of 26 emotion labels. On CAER-S (Table 4), our model

Table 3: Comparison of Category-wise Performance on the EMOTIC Dataset (mAP per Category)

| Category | Kosti et al. Kosti et al. (2017) | Lee et al. Lee et al. (2019) | Zhang et al. Zhang et al. (2019) | Mittal et al. Mittal et al. (2020) | Hoang et al. Hoang et al. (2021) | Yang et al. Yang et al. (2022) | Ours |
|---|---|---|---|---|---|---|---|
| Affection | 26.01 | 22.36 | 46.89 | 45.23 | 44.48 | 37.66 | 52.11 |
| Anger | 11.29 | 12.88 | 10.87 | 15.46 | 30.71 | 17.84 | 32.91 |
| Annoyance | 16.39 | 14.42 | 11.27 | 21.92 | 26.47 | 29.02 | 27.86 |
| Anticipation | 58.99 | 52.85 | 62.64 | 72.12 | 59.89 | 63.31 | 61.79 |
| Aversion | 9.56 | 3.26 | 5.93 | 17.81 | 12.43 | 15.28 | 11.88 |
| Confidence | 81.09 | 72.68 | 72.49 | 68.65 | 79.24 | 74.42 | 80.48 |
| Disapproval | 16.28 | 15.37 | 11.28 | 19.82 | 24.54 | 23.52 | 26.78 |
| Disconnection | 21.25 | 22.01 | 26.91 | 43.12 | 34.24 | 28.95 | 38.79 |
| Disquietment | 20.13 | 10.84 | 16.94 | 18.73 | 24.23 | 21.17 | 24.67 |
| Doubt/Confusion | 33.57 | 26.07 | 18.68 | 35.12 | 25.42 | 24.96 | 29.90 |
| Embarrassment | 3.08 | 1.88 | 1.94 | 14.37 | 4.26 | 10.57 | 3.84 |
| Engagement | 86.27 | 73.71 | 88.56 | 91.12 | 88.71 | 75.23 | 88.16 |
| Esteem | 18.58 | 15.38 | 13.33 | 23.62 | 17.99 | 20.29 | 16.33 |
| Excitement | 78.54 | 70.42 | 71.89 | 83.26 | 74.21 | 86.56 | 75.97 |
| Fatigue | 10.31 | 6.29 | 13.26 | 16.23 | 22.62 | 33.58 | 29.46 |
| Fear | 16.44 | 7.47 | 4.21 | 23.65 | 13.92 | 36.68 | 16.62 |
| Happiness | 55.21 | 53.73 | 73.26 | 74.71 | 83.02 | 85.25 | 86.85 |
| Pain | 10.00 | 8.16 | 6.52 | 13.21 | 16.68 | 19.27 | 29.52 |
| Peace | 22.94 | 19.55 | 32.85 | 34.27 | 28.91 | 26.24 | 32.72 |
| Pleasure | 48.65 | 34.12 | 57.46 | 65.53 | 55.47 | 67.68 | 54.78 |
| Sadness | 19.29 | 17.75 | 25.42 | 23.41 | 42.87 | 47.80 | 54.48 |
| Sensitivity | 8.94 | 6.94 | 5.99 | 8.32 | 15.89 | 24.89 | 18.02 |
| Suffering | 17.60 | 14.85 | 23.39 | 26.39 | 46.23 | 46.74 | 48.76 |
| Surprise | 21.96 | 17.46 | 9.02 | 17.37 | 16.27 | 27.03 | 18.09 |
| Sympathy | 15.25 | 14.89 | 17.53 | 34.28 | 15.37 | 25.87 | 21.40 |
| Yearning | 9.01 | 4.84 | 10.55 | 14.29 | 10.04 | 11.12 | 12.71 |
| Average mAP | 28.33 | 23.85 | 28.42 | 35.48 | 35.16 | 37.73 | 37.88 |

achieves the best accuracy of 88.82. On HECO (Table 5), our model attains the highest recall and F1, suggesting better generalization, while maintaining strong accuracy.

Table 4: Comparison of Performance on the CAER-S Dataset

| Method | Accuracy ↑ |
|---|---|
| Kosti et al. Kosti et al. (2017) | 74.48 |
| Lee et al. Lee et al. (2019) | 73.51 |
| Zhang et al. Zhang et al. (2019) | 77.02 |
| Gao et al. Gao et al. (2021) | 81.31 |
| Li et al. Li et al. (2021) | 74.45 |
| Li et al. Li et al. (2023b) | 84.82 |
| Ours | 88.82 |

Table 5: Comparison of Performance on the HECO Dataset

| Method | Precision ↑ | Recall ↑ | F1 Score ↑ | Accuracy ↑ |
|---|---|---|---|---|
| Kosti et al. Kosti et al. (2017) | 10.20 | 13.88 | 9.89 | 38.80 |
| Lei et al. Lei et al. (2024) Zero-shot | 42.54 | 25.72 | 21.27 | 35.56 |
| Lei et al. Lei et al. (2024) Few-shot | 37.26 | 28.15 | 27.57 | 47.55 |
| Lei et al. Lei et al. (2024) Few-shot CoT | 25.19 | 22.38 | 19.53 | 39.80 |
| Lei et al. Lei et al. (2024) Fine-tune | 48.31 | 33.62 | 35.81 | 60.82 |
| Ours | 41.92 | 44.52 | 42.78 | 57.89 |

## 4.4 Qualitative Results

We provide qualitative predictions on EMOTIC to illustrate the complementarity between our two stages. The Stage 1 VLM tends to produce fewer labels with higher precision, whereas the Stage 2 classifier yields a broader yet accurate label set. Representative examples and figures are moved to Appendix B.

## 4.5 Ablation Study

We conduct several ablation studies to evaluate the effectiveness of various components in our proposed system, including visual feature composition, instruction tuning strategy, loss functions, and the overall impact of incremental enhancements to the VLM. All experiments are conducted on the EMOTIC dataset.

### 4.5.1 Effect of Visual Feature Composition

We first examine how different combinations of visual modules contribute to emotion classification. As shown in Table 6, progressively incorporating more visual cues leads to consistent improvements in mean Average Precision (mAP), indicating that fine-grained information from the human body and face plays a critical role in emotion understanding.

Table 6: Effect of Visual Feature Composition on the EMOTIC Dataset

| Visual Feature Composition | mAP |
|---|---|
| Scene only | 35.51 |
| Scene + Body | 36.79 |
| Scene + Body + Face | 37.88 |

### 4.5.2 Effect of Instruction Tuning Strategy

We explore the impact of instruction tuning by incrementally adding instruction tasks. Table 7 shows that the classification-only setting provides a strong baseline. Adding pose and situation description further boosts performance, and incorporating emotion rationale

understanding yields the best result, demonstrating the importance of diverse instruction prompts in aligning VLM understanding with human emotion reasoning.

Table 7: Effect of Instruction Tuning Strategies on the EMOTIC Dataset

| Instruction Tuning Strategy | mAP |
| --- | --- |
| Classification task | 36.91 |
| + Pose and situation description | 37.34 |
| + Emotion rationale task | 37.88 |

### 4.5.3 Effect of Data Balancing Strategy

We evaluate the effectiveness of our data balancing strategy by comparing model performance with and without the balancing mechanism. As shown in Table 8, our strategy yields a slight improvement in overall mAP, indicating enhanced robustness across the dataset.

To further assess its impact on minority categories, we report results for the three least frequent emotion labels—Embarrassment, Aversion, and Fear. As shown in Table 9, data balancing improves performance for these underrepresented classes.

Table 8: Effect of Data Balancing Strategies on the EMOTIC Dataset

| Data Balancing Strategy | mAP |
| --- | --- |
| w/o Data Balancing | 37.72 |
| w/ Data Balancing | 37.88 |

Table 9: Performance on the Three Most Minority Labels

| Category | w/o Balancing | w/ Balancing |
| --- | --- | --- |
| Embarrassment | 3.14 | 3.84 |
| Aversion | 10.56 | 11.88 |
| Fear | 13.74 | 16.62 |

## 5 Conclusion

We present a simple two-stage recipe for context-aware emotion recognition: an instruction-tuned VLM to align visual cues with emotion semantics, followed by a lightweight classifier over scene, body, and face features. Across EMOTIC, CAER-S, and HECO, this pairing matches or surpasses prior work that relies on heavier feature stacks, while remaining compute-friendly and reproducible. Our label-aware balancing and instruction-tuning corpus further improve recall on minority emotions without sacrificing precision.

Limitations include reliance on VLM quality and person localization; extending to end-to-end training, temporal cues, and robustness under occlusion and long-tail shifts are promising directions.

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

# A Dataset

In this chapter, we introduce the context-aware emotion recognition datasets used in this study, along with a detailed analysis of their characteristics. We also describe the construction process of our QA dataset, including strategies for addressing data imbalance and applying instruction tuning. Finally, we present the overall dataset statistics.

## A.1 Context-Aware Emotion Datasets

We provide a comprehensive overview of various context-aware emotion recognition datasets, including both image-based and video-based datasets, as summarized in Table 10. Although video-based datasets are listed for completeness, we focus our analysis on image-based datasets and their label distributions.

Table 10: Overview of Context-Aware Emotion Recognition Datasets

| Dataset | Modality | Dataset Size | Agents Annotated | Source | Labels |
|---|---|---|---|---|---|
| EMOTIC Kosti et al. (2017) | Images | 23,189 images | 33,783 | Web | 26 Categories (Multi-label) |
| CAER-S Lee et al. (2019) | Images | 70,000 images | 70,000 | TV Shows | 7 Categories |
| HECO Yang et al. (2022) | Images | 9,385 images | 19,781 | Web | 8 Categories |
| CAER Lee et al. (2019) | Videos | 13,201 clips | 13,201 | TV Shows | 7 Categories |
| IEMOCAP Busso et al. (2008) | Videos | 12 hours | - | TV Shows | 4 Categories |
| GroupWalk Mittal et al. (2020) | Videos | 45 clips | 3,544 | Real-world | 4 Categories |

### A.1.1 EMOTIC

The EMOTIC dataset Kosti et al. (2017) is composed of images collected from three sources: MSCOCO Lin et al. (2015), ADE20K Zhou et al. (2019), and additional images manually gathered using the Google search engine. All images are annotated by qualified human annotators. Figure 3 presents sample images and their emotion annotations.

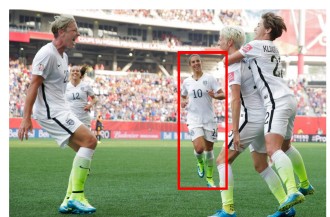 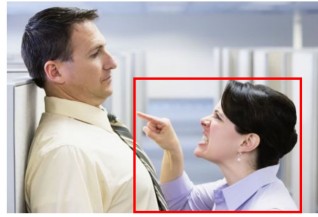 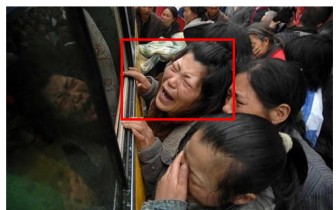

Affection, Confidence, Engagement, Excitement, Happiness, Pleasure

Anger, Annoyance, Disapproval, Suffering

Pain, Suffering, Sadness, Disapproval

Figure 3: Example images and annotations from the EMOTIC dataset.

Each agent in the dataset is labeled with one or more of 26 discrete emotion categories in a multi-label format. These categories are: Engagement, Happiness, Anticipation, Excitement, Confidence, Pleasure, Peace, Disconnection, Affection, Esteem, Sympathy, Yearning, Doubt/Confusion, Fatigue, Disquietment, Surprise, Sadness, Annoyance, Sensitivity, Disapproval, Suffering, Anger, Pain, Fear, Aversion, and Embarrassment.

The final version of the dataset consists of 23,189 images and 33,783 annotated human agents. It is split into training (70%), validation (10%), and testing (20%) subsets. The training set is annotated by a single annotator per agent, while the validation and test sets are labeled by five and three annotators, respectively, to ensure more reliable evaluation.

### A.1.2 Data Distribution

A total of 65,889 emotion labels are assigned to 26,581 annotated agents in the EMOTIC training and validation sets, resulting in an average of 2.48 labels per sample.

As shown in Figure 4, the distribution exhibits a clear long-tail pattern. The top five most frequent emotion categories—Engagement (24.20%), Anticipation (12.59%), Happiness (12.00%), Excitement (9.65%), and Confidence (9.03%)—collectively make up over 67% of all annotations. In contrast, 10 of the 26 emotion categories each comprise less than 1% of the total labels. Among the rarest labels are Fear (0.53%), Anger (0.48%), and Embarrassment (0.47%).

Notably, the most dominant label, Engagement, accounts for 15,947 instances (24.20%), indicating a substantial imbalance in the label distribution. In addition, Engagement appears as the sole label in 5,112 samples, further highlighting its dominance, which may lead to biased model learning.

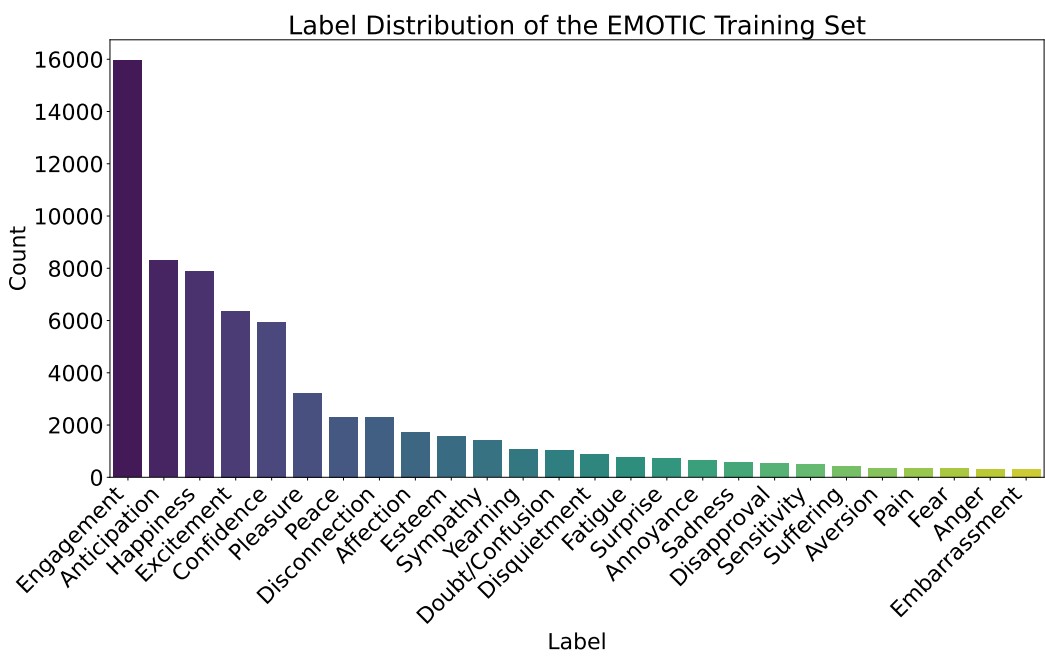

Figure 4: Label distribution of the EMOTIC training set. The top 5 most frequent emotion categories— Engagement (24.20%), Anticipation (12.59%), Happiness (12.00%), Excitement (9.65%), and Confidence (9.03%)—account for over 67% of the annotations, indicating a long-tail distribution and significant class imbalance.

### A.1.3 CAER-S

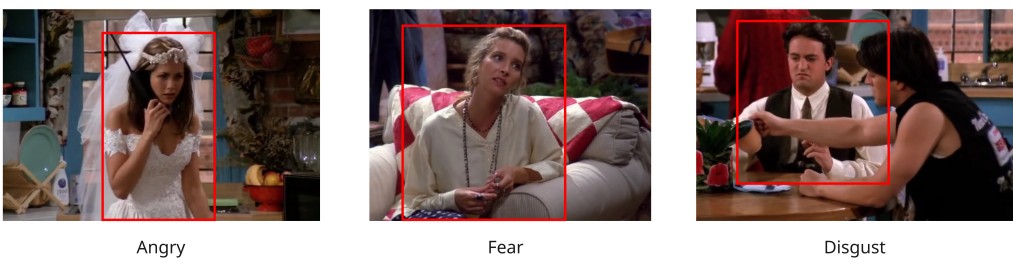

Angry       Fear       Disgust

Figure 5: Example images and annotations from the CAER-S dataset.

The CAER-S dataset consists of 70,000 static images extracted from 79 TV shows. It is a subset of the CAER dataset Lee et al. (2019). To construct the CAER dataset, video clips in TV shows are processed by shot boundary detector, face detector and feature clustering.

Each resulting video clip was then annotated with one of seven emotion categories: Anger, Disgust, Fear, Happy, Sad, Surprise, and Neutral, by a group of six annotators. Figure 5 presents sample images and their emotion annotations.

The 70,000 images are evenly distributed across the seven emotion categories, with 10,000 images per class, ensuring a well-balanced dataset. The dataset is randomly split into training (70%), validation (10%), and testing (20%) subsets.

### A.1.4 HECO

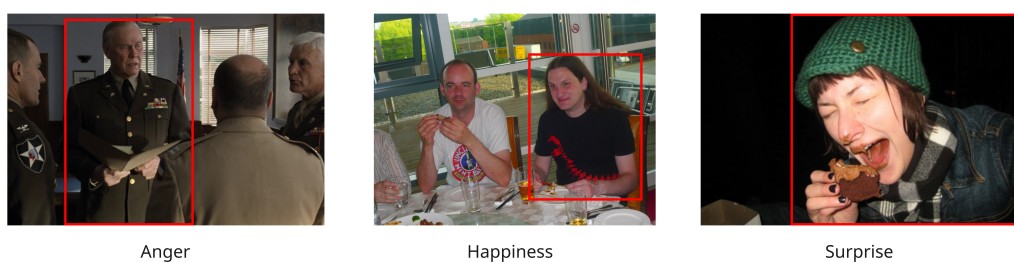

Anger            Happiness            Surprise

Figure 6: Example images and annotations from the HECO dataset.

The HECO dataset, introduced by Yang et al. (2022), contains 9,385 images and 19,781 annotated human agents. Images in the dataset are collected from three sources: HOI datasets Chao et al. (2018), film clips, and web images. Figure 6 presents sample images and their emotion annotations.

The annotation process was conducted by 3 professional psychologists and 10 graduate students. Each agent in the image is labeled with one of 8 discrete emotion categories: Surprise, Excitement, Happiness, Peace, Disgust, Anger, Fear, and Sadness.

To improve robustness, approximately 2% of the dataset consists of intentionally fuzzy images, and 5% of the images contain agents affected by occlusion.

### A.1.5 Data Distribution

The HECO dataset contains a total of 19,781 labeled instances across 8 discrete emotion categories. As shown in Figure 7, the dataset exhibits a moderate class imbalance.

The most frequent emotion is Happiness, with 7,113 instances (35.96%), followed by Peace (5,886; 29.76%). Together, these two categories account for over 65% of the entire dataset. In contrast, low-frequency emotions such as Sadness (3.55%), Surprise (3.86%), and Anger (4.27%) are significantly underrepresented.

### A.2 QA Dataset Preparation

### A.2.1 Task Formulation and Prompt Design

We formulate the problem as a classification task in the form of a vision-language QA prompt. The model is asked to infer a set of likely emotions from a fixed list of candidates based on the image and textual description. The base QA prompt format is:

Based on the image, please select multiple emotions that the person
likely contains from the following options: Embarrassment, Aversion,
Fear, Pain, Anger, Suffering, Disapproval, Sensitivity, Annoyance,
Sadness, Surprise, Disquietment, Fatigue, Doubt/Confusion, Yearning,
Sympathy, Esteem, Affection, Disconnection, Peace,  Pleasure,
Confidence, Excitement, Anticipation, Happiness, Engagement.

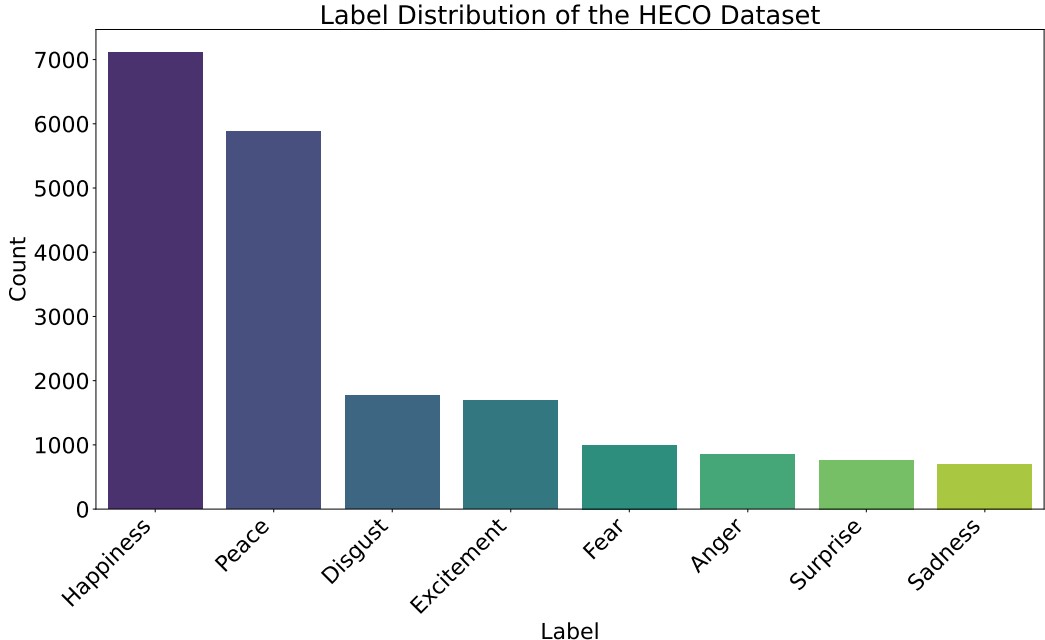

Figure 7: Label distribution of the HECO Dataset. The top 2 most frequent emotion categories— Happiness (35.96%), and Confidence (29.76%)—account for over 65% of the annotations, indicating a significant class imbalance.

### A.2.2 Data Balancing Strategy

The long-tail nature of the label distribution poses challenges for model training, as models tend to be biased toward dominant categories while underperforming on rare ones. To address this issue, we explore techniques such as data augmentation, and sampling methods.

To address the severe label imbalance in emotion datasets such as EMOTIC and HECO, we propose a label-aware data balancing strategy. This strategy improves model robustness in recognizing minority emotion categories by combining minor-label QA augmentation with major-label downsampling.

### A.2.3 Minor-Label QA Augmentation

To increase exposure to minority emotions, we generate an additional QA sample for any data point that includes at least one minor label. In this augmented version, only the minor labels are included in the candidate answer list, and major labels are excluded to prevent dominant class interference. The augmented prompt format is:

Based on the description and image, please select multiple emotions
that the person likely contains from the following options:
Embarrassment, Aversion, Fear, Pain, Anger, Suffering, Disapproval,
Sensitivity, Annoyance, Sadness, Surprise, Disquietment, Fatigue,
Doubt/Confusion, Yearning, Sympathy, Esteem, Affection, Disconnection,
Peace.

### A.2.4 Major-Label Down Sampling

To further alleviate imbalance, we down-sample training samples that contain only major emotion labels. We define the major labels based on their high frequency in each dataset:

> [nosep, leftmargin=2em]EMOTIC: Engagement, Happiness, Anticipation, Excitement, Confidence HECO: Happiness, Peace

We apply the following dataset-specific sampling ratios:

> [nosep, leftmargin=2em]For EMOTIC, samples containing only major labels are down-sampled at a rate of 0.2. For HECO, the down-sampling rate is 0.5. Additionally, in EMOTIC, samples that only have the single label Engagement (comprising over 20% of training samples) are down-sampled at a rate of 0.8, reducing this dominant subset.

## A.3   Instruction Tuning Dataset Generation

To enable the Q-Former and LLM to handle diverse instruction-following tasks across different formats, we construct an instruction-tuning dataset that simulates various types of reasoning required for visual emotion understanding. Compared to multi-task learning, instruction tuning offers a unified training paradigm that encourages better generalization across emotion-related tasks.

Given the high cost of manual annotation, we adopt an automatic data generation pipeline, leveraging state-of-the-art vision-language models (VLMs) to synthesize high-quality question-answer (QA) pairs.

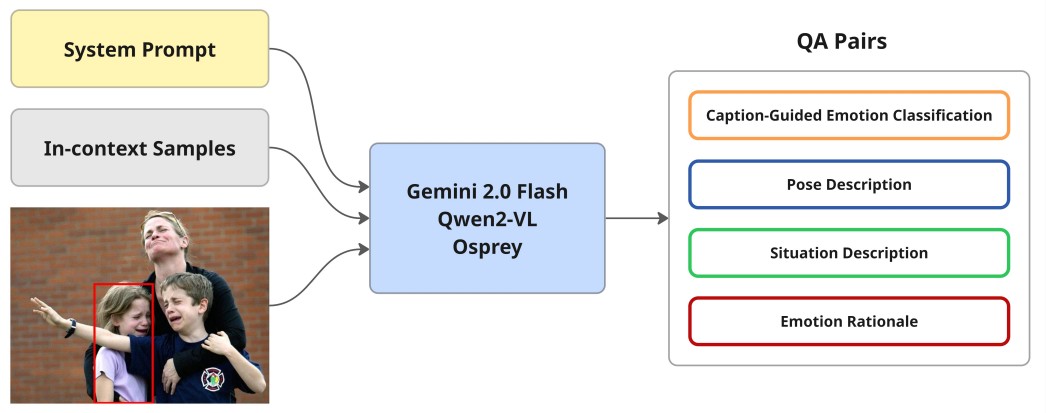

Figure 8:   Overview of our instruction-tuning dataset generation pipeline.

Figure 8 illustrates our instruction-tuning data generation pipeline. The resulting dataset comprises various types of instruction-following samples, including person-centric captions, pose and situation descriptions, and rationale-based explanations. Each QA pair is explicitly grounded on a target individual, indicated by a red bounding box in the image.

**Caption-Guided Emotion Classification**   We use the Osprey-7B Yuan et al. (2024) to generate concise, person-centric captions focused on the target agent's appearance and actions. Osprey-7B supports input masks that allow it to describe specific regions in an image, which enables us to create emotionally relevant captions conditioned on the target person. These captions are then used as context for emotion inference tasks.

**Visual Description**   We employ Qwen2-VL-7B Bai et al. (2023) in an in-context few-shot learning setting to produce pose and situation descriptions. These QA pairs provide contextual cues that go beyond facial expression, incorporating body posture, surrounding elements, and interactions with other people. The red bounding box guides the model to describe only the specified agent.

**Emotion Rationale**   We employ Gemini 2.0 Flash Team (2025) to generate reasoning-style QA pairs, where the model explains the emotional state of the target person using a chain-of-thought rationale. These QA pairs guide the VLM to learn interpretable reasoning processes by incorporating visual cues, interpersonal context, and emotional expressions, ultimately enhancing its ability to understand and predict emotions more effectively.

Prompt templates for all tasks can be found in the Appendix. And dataset examples can be found in Figure 12

## A.4 Dataset Statistics after Processing

### A.4.1 Label Distribution Before and After Balancing

Since CAER-S is a balanced dataset, we only apply balancing strategies to the EMOTIC and HECO datasets. The label counts are computed based on their occurrence in the classification task answers.

EMOTIC Dataset   As shown in Figure 9, the dominant label Engagement slightly decreases in frequency after preprocessing due to downsampling. In contrast, many minority labels, such as Fear, Embarrassment, and Aversion, exhibit noticeable increases, demonstrating the effectiveness of our augmentation strategies.

Overall, the label distribution becomes more balanced, with the top label's proportion reduced from 24.20% to 14.49%. However, due to the severe original imbalance, long-tail issues still persist and will be further mitigated using loss-based techniques in training.

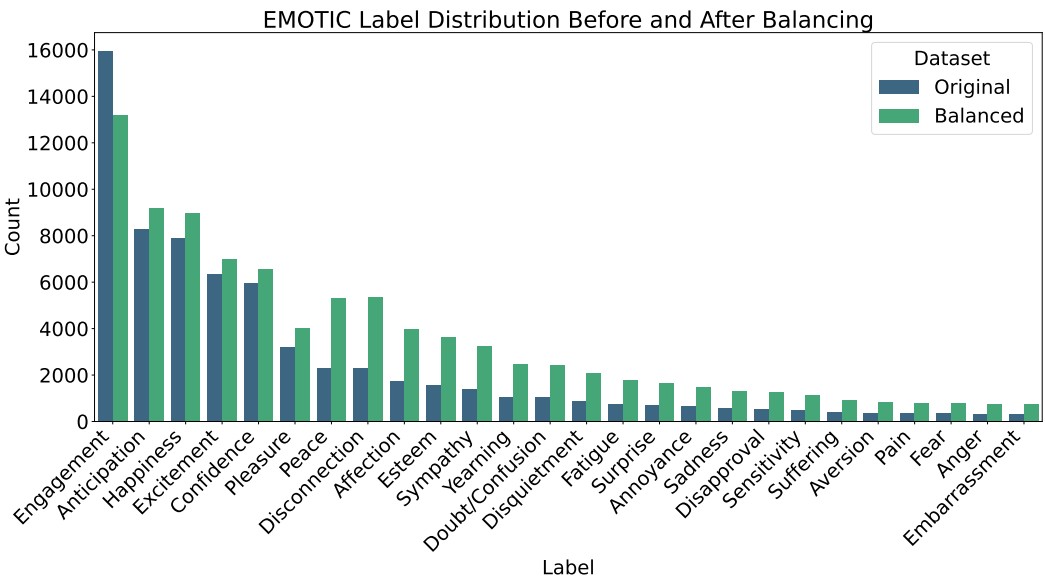

Figure 9:   Label distribution of the EMOTIC dataset before and after preprocessing.

HECO Dataset   The HECO dataset is a multi-class dataset, where each sample is associated with a single emotion label, making it less susceptible to extreme label imbalance compared to multi-label datasets. After applying our preprocessing pipeline, the distribution becomes notably more uniform, as shown in Figure 10.

### A.4.2 Instruction-Tuning Task Composition

We analyze the composition of our instruction-tuning dataset using the EMOTIC dataset as an example. To prevent classification performance from being overshadowed by other task types, we limit the number of non-classification tasks. This allows the Q-former to learn generalized features while still supporting strong classification performance. Figure 11 illustrates the distribution of task types in the final instruction-tuning dataset.

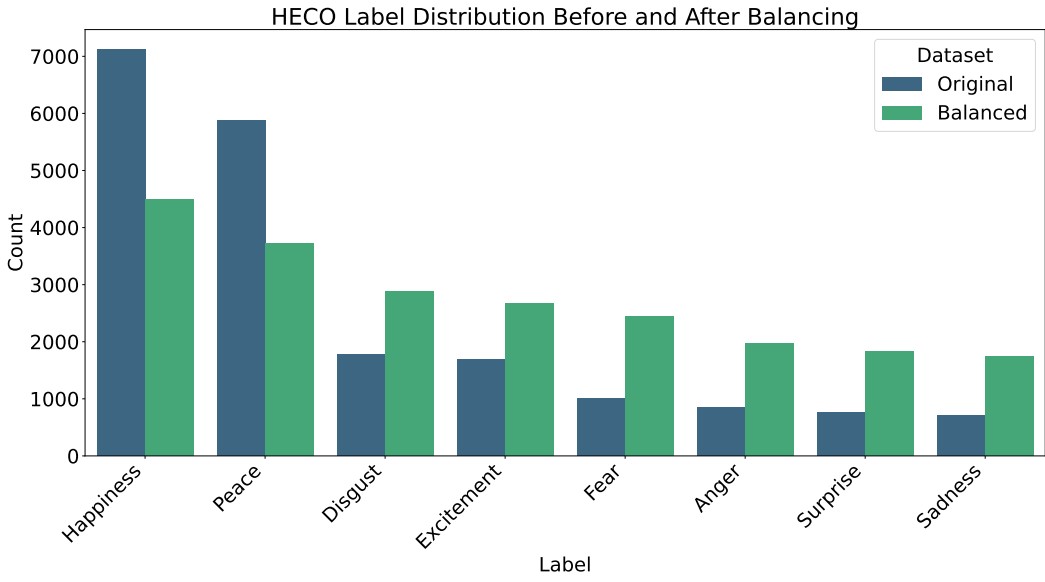

Figure 10: Label distribution of the HECO dataset before and after preprocessing.

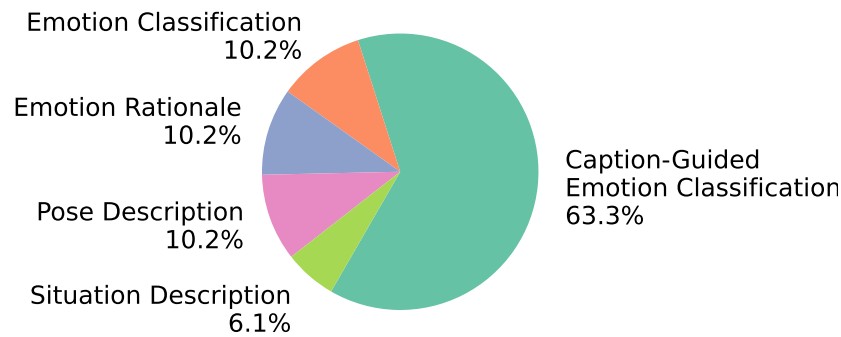

Figure 11: Distribution of task types in the instruction-tuning dataset constructed from the EMOTIC dataset.

# B  Additional Qualitative Examples

Figure 13. In the figure, the target agent is highlighted with a red bounding box. We show the model predictions from both the Stage 1 VLM model and the Stage 2 classifier.

## Emotion Instruction Tuning Data Example

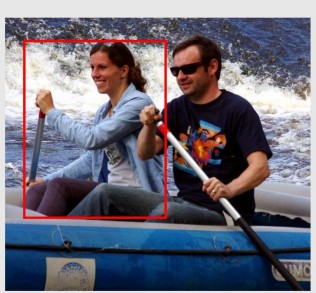

**Emotion Category**
Esteem, Affection, Pleasure, Excitement, Happiness

### Emotion Classification

**Question:** Based on the image, please select multiple emotions that the person likely contains from the following options: Embarrassment, Aversion, Fear, Pain, Anger, Suffering, Disapproval, Sensitivity, Annoyance, Sadness, Surprise, Disquietment, Fatigue, Doubt/Confusion, Yearning, Sympathy, Esteem, Affection, Disconnection, Peace, Pleasure, Confidence, Excitement, Anticipation, Happiness, Engagement.
**Answer:** Esteem, Affection, Pleasure, Excitement, Happiness.

### Caption-Based Classification

**Question:** In the image, **A woman is seen on the left side of the image, wearing a blue shirt and a white skirt. She is holding an oar and appears to be paddling a boat.** Based on the description and image, please select multiple emotions that the person likely contains from the following options: Embarrassment, Aversion, Fear, Pain, Anger, Suffering, Disapproval, Sensitivity, Annoyance, Sadness, Surprise, Disquietment, Fatigue, Doubt/Confusion, Yearning, Sympathy, Esteem, Affection, Disconnection, Peace, Pleasure, Confidence, Excitement, Anticipation, Happiness, Engagement.
**Answer:** Esteem, Affection, Pleasure, Excitement, Happiness.

### Pose Description

**Question:** How would you describe the body posture of the person of the image?
**Answer:** The person is a woman with brown hair in a low ponytail. She is wearing a light blue jacket, a white shirt, and purple pants. She is sitting on a raft, facing the water, and holding a paddle in her right hand. Her left arm is bent at the elbow and held behind her body. The woman is smiling as she gazes toward the water behind her. The water appears to be crashing against the raft.

### Situation Description

**Question:** What situation is depicted in the image?
**Answer:** The scene takes place on a choppy body of water, possibly a river or an ocean, where two individuals are navigating a small raft. The man and woman both appear to be actively engaged in paddling, gripping their paddles firmly. The water's movement is pronounced, suggesting that they might be encountering waves or a strong current. The environment around them appears naturally vast and slightly turbulent, indicating an outdoor adventure or recreational activity rather than a leisurely cruise. The situation involves the man and woman collaborating to propel the raft, each holding a paddle.

### Emotion Rationale

**Question:** Why is the person feeling this way?
**Answer:** The woman is smiling widely, suggesting positive emotions. She is participating in a recreational activity (rowing), which often leads to feelings of enjoyment. Her engaged posture and facial expression further indicate pleasure and excitement. The overall context of being on the water, possibly with a companion, points towards affection and happiness. Her focused expression might indicate esteem towards her achievement of the activity.

Figure 12: The sample of our generated visual emotion instruction tuning dataset

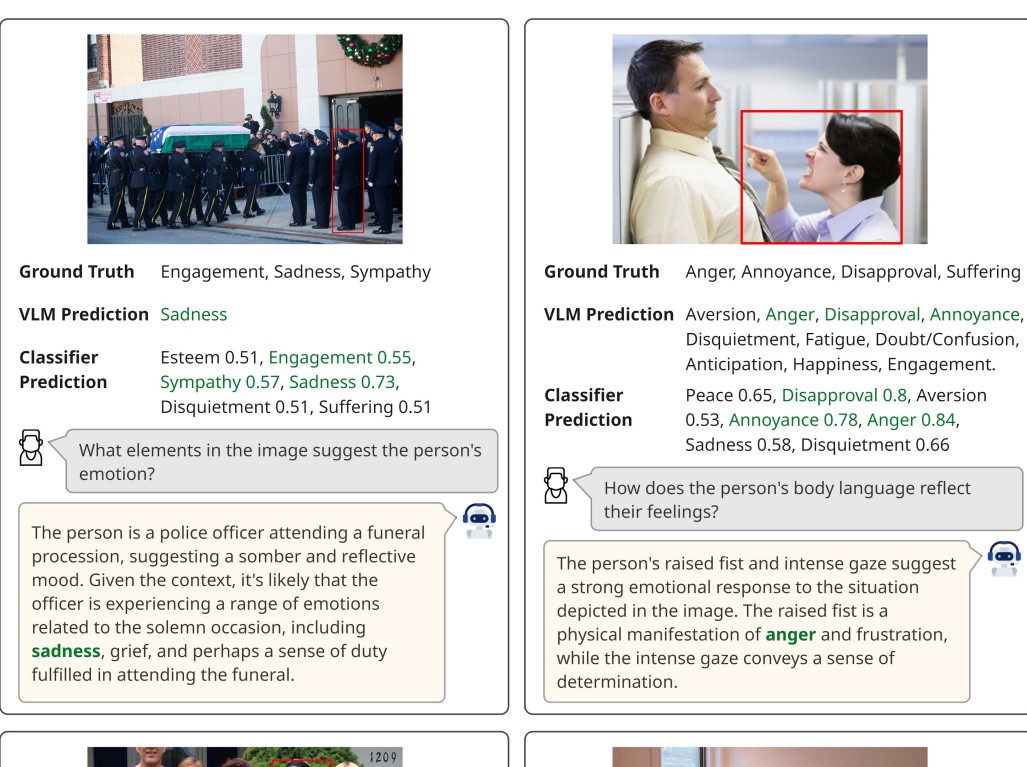

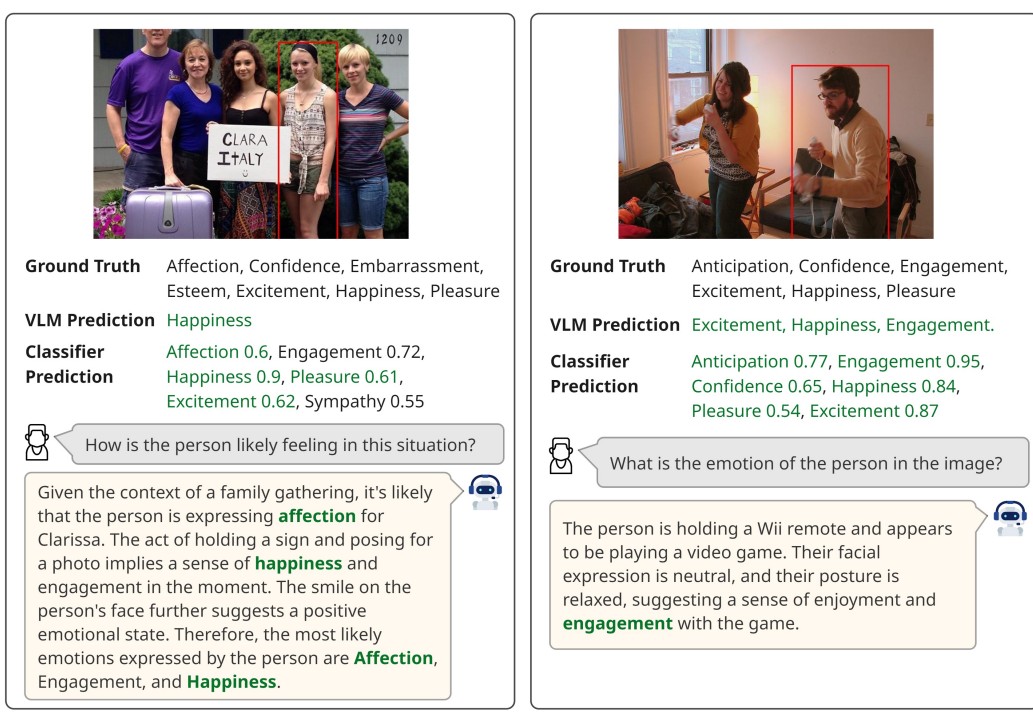

Figure 13: Qualitative Result on the EMOTIC dataset.

The VLM tends to predict fewer emotion labels, but with higher precision. In contrast, our classifier outputs a broader set of emotion labels without sacrificing accuracy. Notably, predictions with higher confidence scores tend to be more accurate.

We also demonstrate the generalization ability of our VLM: users can ask emotion-related questions, and the model is capable of providing reasoned responses along with precise emotion labels.

## C   More Ablation Study

### C.1   Effect of Self-Attention Mechanism

To evaluate the contribution of the self-attention layer in our classifier architecture, we conduct an ablation study by removing the self-attention block and directly concatenating the features before classification. As shown in Table 11, removing the self-attention results in a noticeable performance drop. This demonstrates that the self-attention mechanism plays a crucial role in capturing the inter-modality relationships among the face, body, and scene features.

Table 11: Effect of Self-Attention Mechanism on the EMOTIC Dataset

| Architecture | mAP |
| --- | --- |
| w/o Self-Attention | 37.11 |
| w/ Self-Attention | 37.88 |

### C.2   Effect of Loss Function

We compare two loss functions for multi-label classification: binary cross entropy and asymmetric loss. As shown in Table 12, asymmetric loss significantly outperforms binary cross entropy, highlighting its suitability for imbalanced emotion label distributions.

Table 12: Effect of Different Loss Functions on the EMOTIC Dataset

| Loss Function | mAP |
| --- | --- |
| Binary Cross Entropy | 33.94 |
| Asymmetric Loss | 37.88 |

### C.3   Effect of Incremental VLM Enhancements

Lastly, we present the performance impact of incrementally enhancing our VLM-based architecture, as detailed in Table 13. Starting from a baseline using only scene information and simple classification prompts, we observe that the introduction of caption-guided prompts and body-level features gradually improves the F1 score. Instruction tuning and data balancing strategies significantly increase recall. The final integration of the face module and advanced instruction tuning further boosts precision and reduces Hamming loss.

Table 13: Performance Impact of Incremental VLM Enhancements on the EMOTIC Dataset

| Configuration | Precision ↑ | Recall ↑ | F1 Score ↑ | Hamming Loss ↓ |
| --- | --- | --- | --- | --- |
| Category classification (Scene only) | 32.83 | 33.08 | 29.50 | 17.59 |
| + Caption-guided prompt | 35.88 | 34.30 | 31.43 | 16.54 |
| + Person module | 30.70 | 35.01 | 31.88 | 16.54 |
| + Data balancing and instruction tuning | 33.58 | 40.78 | 33.65 | 18.66 |
| + Face module | 48.72 | 18.82 | 22.97 | 13.57 |
| + Final instruction tuning | 43.41 | 18.92 | 21.58 | 13.21 |

# D    Dataset Construction

## D.1    Pose Description Task

We design a prompt with in-context examples to generate pose descriptions focused on the target agent, as shown in Listing 1. Subsequently, we apply the instruction template in Listing 2 to construct corresponding question-answer pairs.

---

You are an expert assistant for generating structured pose descriptions from images. Your task
   is to analyze an image with a target person inside a red bounding box and describe their
   pose in a concise, structured manner.

**Guidelines:**
- Only describe the person inside the red bounding box and do not mention "red bounding box"
   or "red rectangle" in the description.
- Keep the description brief (one to two sentences).
- Mention the head position, body posture, leg position, arm position and hand gestures (if
   visible).
- Do not mention emotions and face expressions.
- Use consistent and clear wording.

### **Examples:**

#### **Example 1**
**Input:**
- Image with a red bounding box.

**Output:**
"A man stands upright, facing forward with arms crossed over his chest. He maintains direct
   eye contact."

#### **Example 2**
**Input:**
- Image with a red bounding box.

**Output:**
"A woman jogs forward with her head slightly turned to the left. Her hands are open and
   relaxed."

### Question
**Input:**
- Image with a red bounding box.

**Output:**

---

Listing 1: Prompt Template for Generating Pose Description

```
- How would you describe the pose of the person in the image?
- How would you describe the body posture of the person in the image?
- Can you describe the stance and gestures of the person located?
- Describe the physical stance and posture of the person.
- What is the body language of the individual?
- Can you explain how the person is holding their body?
- What are the notable aspects of the person's movement and posture?
- What details can you provide about the orientation and posture of the person?
- What kind of physical expression is visible in the person's pose?
- How does the person's pose reflect their activity?
- What are the defining characteristics of the person's pose?
- What is the overall posture of the person?
- What do the person's posture and gestures suggest about their action?
- How would you characterize the person's body dynamics?
- What kind of motion, if any, is suggested by the person's stance?
```

Listing 2: Pose Instruction Template

## D.2 Situation Description Task

We design a prompt with in-context examples to generate situation descriptions from the target agent's perspective, as shown in Listing 3. Subsequently, we apply the instruction template in Listing 4 to construct corresponding question-answer pairs.

```
You are an AI model trained to analyze images and provide a detailed background and
    situation description for the person in the region (inside the red bounding box). Your task is
    to describe:
1. The background: General environmental context (e.g., street, park, office).
2. The situation: The activity or event happening in the scene, but only including information
    relevant to the target person.
Do not describe the target person's appearance, clothing, posture, or facial expressions.

## Examples
### Example 1
Input:
(Image of a young man in a red bounding box sitting at a desk in an office with coworkers in
    the background.)

Output:
The scene takes place in a modern office environment with multiple desks and employees
    working on their computers. The atmosphere suggests a professional setting where people
    are engaged in various tasks. The person in the region appears to be focused on work,
    possibly handling documents or writing emails as part of a typical office routine.

### Example 2
Input:
(Image of a woman in a red bounding box jogging in a park with trees and other joggers.)

Output:
The setting is a public park with lush greenery, a paved jogging path, and a few other people
    exercising. The environment suggests a peaceful and refreshing outdoor space, commonly
    used for fitness activities. The person in the region is actively participating in exercise,
    moving along the path in sync with other joggers, likely as part of a workout or daily
    routine.

<image>
Now, this is the image you need to analyze.
Please generate a description following the format of the examples above. Ensure that the
    description includes details about the background and situation while staying relevant to the
    individual in the region (red box).
```

Listing 3: Prompt Template for Generating Situation Description

```
- What is the setting of the scene?
- Describe the environment surrounding the person.
- What is happening in the background?
- What is happening in the image, and what does it suggest about the context or event?
- Describe the situation involving the person. What might they be experiencing or doing?
- What kind of event or moment is being captured?
- What situation is depicted in the image?
- What is the context of the scene?
- What is the background of the image?
- What is the environment like?
- Describe the situation in the image.
```

Listing 4: Situation Instruction Template

## D.3 Rationale Task

We design a prompt with in-context examples to generate emotion rationale focusing on the target agent and ground-truth labels, as shown in Listing 5. Subsequently, we apply the instruction template in Listing 6 to construct corresponding question-answer pairs.

```
You are given an image with a specific region (red box) containing a person. Your task is to
    generate a detailed question-answer pair that helps a Vision Language Model (VLM) learn
    how to associate emotions with various contextual cues.

Task:
1. Generate one high-quality QA pair per image to encourage deep reasoning.
2. Ensure the answer is structured, insightful, and logically analyzes the image and visual cues
    before answering.
3. The answer should be concise, limited to 5 sentences.
4. If some contextual cues are unclear, focus on the available ones.
5. Output should be in a structured JSON format for easy parsing.

The question should always follow this format:
"Given the following emotions: {', '.join(labels)}, which emotions are being expressed by the
    person in the region? Please analyze the image, pose, people interaction, and face before
    answering."

Output Example:
[
{
  "question": "Given the following emotions: ..., which emotions are being expressed by the
    person in the region? Please analyze the image, pose, people interaction, and face before
    answering.",
  "answer": "The person in the image is looking at a dog, which is a common activity that can
    evoke feelings of happiness and joy. He is smiling, which indicates a positive emotional state.
     The background shows a sunny day, which can also contribute to a cheerful mood.
    Therefore, the emotions expressed by the person in the region are likely to be happiness."
}
]

The ground truth emotions for the person in the region are {', '.join(gt_labels)}. Do not
    provide answer outside the ground truth.
```

Listing 5: Prompt Template for Generating Emotion Rationale

```
- Why is the person feeling this way?
- What is the emotion expressed by the person?
- What is the emotion of the person?
- What is the emotion of the person in the image?
- Describe the emotion of the person in the image.
- What does the person feel?
- What emotion is the person expressing?
```

```
- Describe details about the emotion of the person.
- What emotion is conveyed through the person's pose or expression?
- What elements in the image suggest the person's emotion?
- What body language or facial features indicate the person's feelings?
- Describe the emotional cues shown by the person.
- How is the person likely feeling in this situation?
- What emotion can be inferred from the person's expression?
- How does the person's body language reflect their feelings?
- What feeling does the person seem to be experiencing?
- How does the person's expression convey their emotion?
- What emotion is the person likely experiencing?
- Describe the emotional state of the person.
```

Listing 6: Rationale Instruction Template

## E  More Qualitative Results

We present additional qualitative results to further demonstrate the effectiveness of our approach. Figure 14 shows the results on the EMOTIC dataset, while Figure 15 shows the results on the HECO dataset.

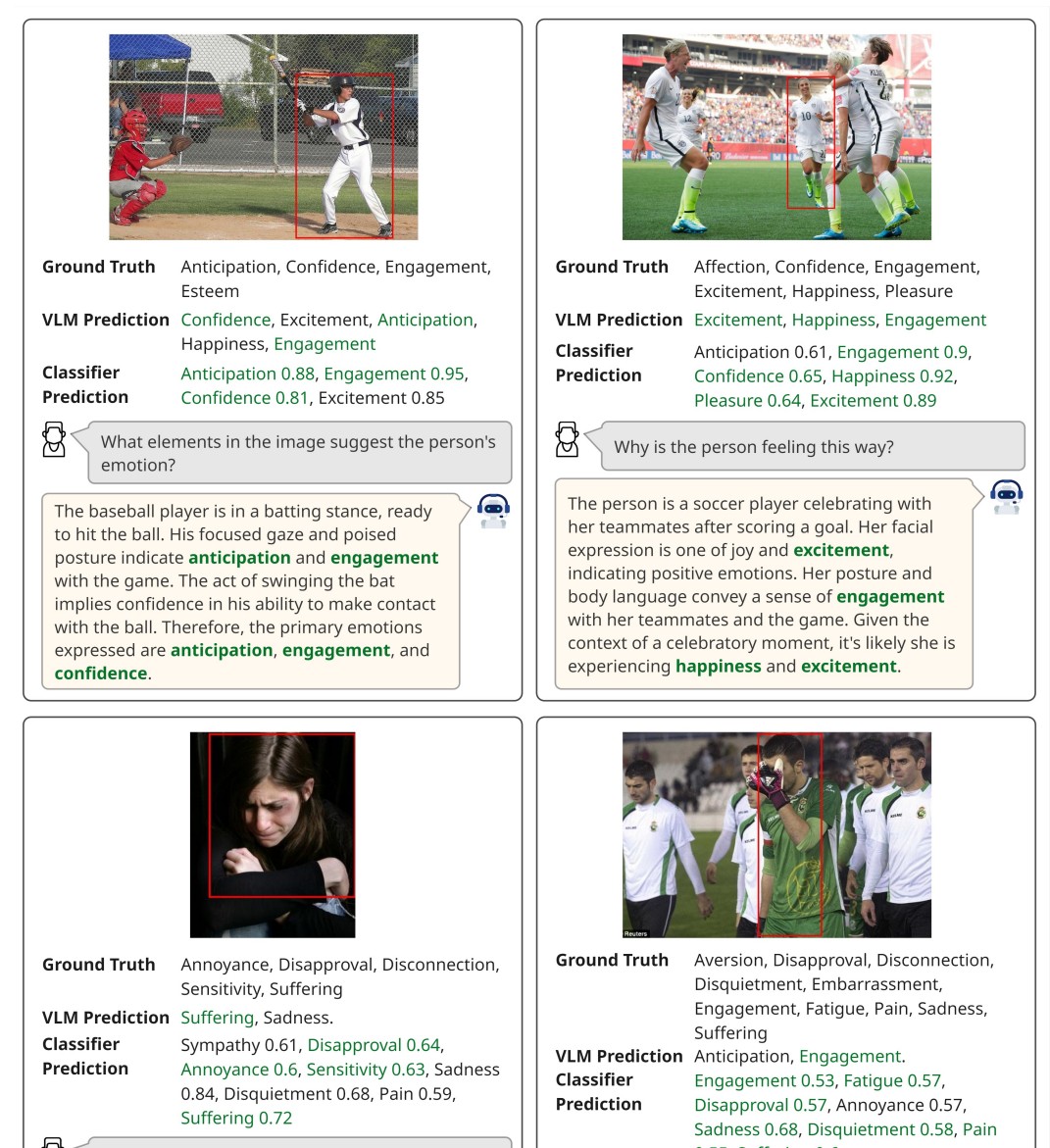

Figure 14:  More Qualitative Results on the EMOTIC Dataset.

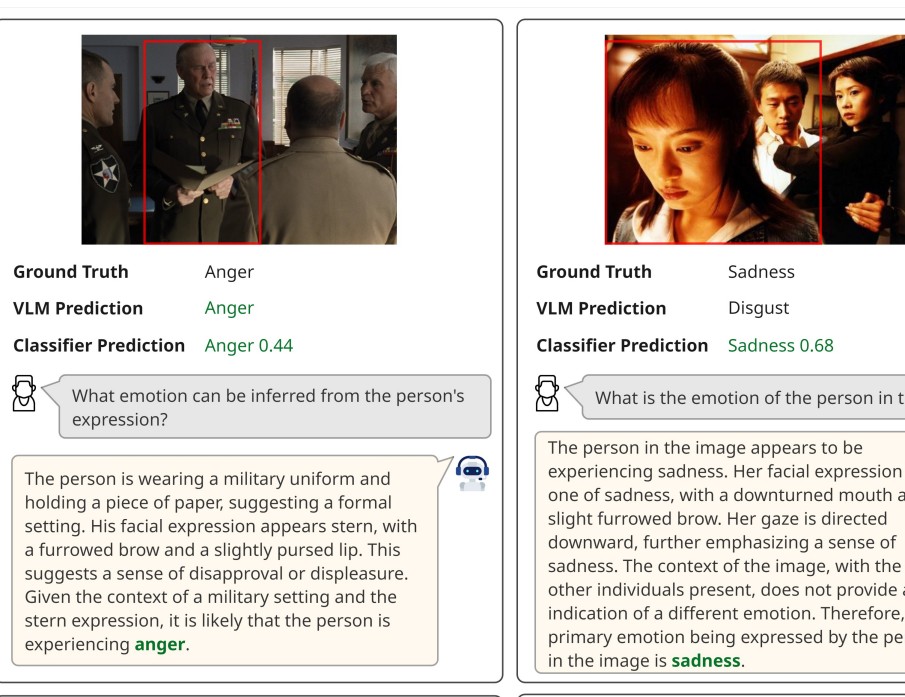

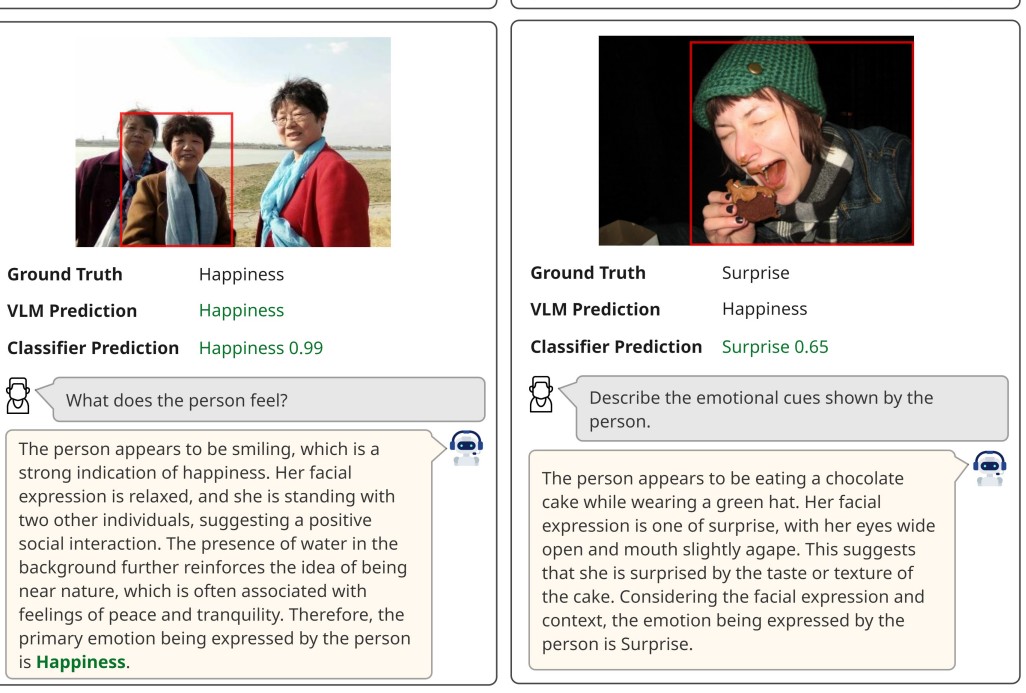

Figure 15: More Qualitative Results on the HECO Dataset.

