# OpenReview forum: "Context-Aware Emotion Recognition via Multi-View  Instruction-Tuned Visual Language Guidance"
_ICLR.cc/2026/Conference — ICLR 2026 Conference Withdrawn Submission_

### Official Review · Reviewer_omPa · 2025-10-23

**Soundness:** 2
**Presentation:** 2
**Contribution:** 2
**Rating:** 2
**Confidence:** 5

**Summary:**

The paper proposes a two-stage method for in-context emotion recognition leveraging a pre-trained VLM. In the first stage, IntructBLIP is finetuned using three separate vision encoders (for the entire image, face and body, respectively). The output is the class names. In the second stage, a classifier using a QFormer to fuse the different streams of the face encoders is further trained on the final classification. The method achieves results lower than those of other SOTA methods, which are unfortunately not included in the paper (see weaknesses).

**Strengths:**

The idea of using three separate vision encoders is intuitive as a method to disambiguate context from subject. In addition, addressing the class imbalance in EMOTIC makes sense as some of the minority classes represent less than 5% of the dataset.

**Weaknesses:**

The biggest weakness is the fact that the related work has been cherry-picked, and key comparisons are not included [1,2,3]. The method is nearly identical to [1], the main difference being the three encoders and two-stage training. The similarities in the method are too many to ignore, specifically the use of VLLMs and finetuning of InstructBLIP for the task. Furthermore, this work performs significantly worse than all [1,2,3] on both EMOTIC and CAER-S, so it is hard to justify the method (more complex, less effective and with limited novelty).

In terms of clarity, the paper has a few issues, one of them being Table 7. There are two rows for" Pose and situation description", and "Emotion rationale task" -- these are not explained in the method, as far as we can see from the figures and the text. In the first stage, the input is 3 RGB images and an instruction in the form of multiple-choice VQA, while the output is class names. Similarly, in the second stage, there is no text input, and the output is standard classification probabilities. Either the method or the experiments need to be updated for clarity. On some more minor clarity/presentation issues, Table 3 is non-legible, and all tables have issues with citations being repeated.

In terms of results significance, Tables 8 & 9 show very similar results to non-balancing, so I am wondering if 0.16% is enough of a methodological contribution for only three classes.

In terms of the text, it looks like the citations in the first paragraph of the intro are not appropriate. Specifically, "[...] early research focused on isolated facial expressions (eg, Ntizikra et al, 2025)" -> a) this is from 2025, so not particularly early, and b) the paper is about IOT [4]. In addition, "The advent of VLMs [..] leveraging their powerful semantic reasoning capabilities (Kosti 2017)" -> That's the EMOTIC dataset.

[1]  Xenos et al. "VLLMs Provide Better Context for Emotion Understanding Through Common Sense Reasoning", 2024

[2] D. Yang, et al. Context De-Confounded Emotion Recognition. CVPR, 2023

[3] D. Yang, et al. Robust emotion recognition in context debiasing. CVPR, 2024

[4] Ntizikira et al. Enhancing IoT security through emotion recognition and blockchain-driven intrusion prevention, 2025

**Questions:**

Further to fixing the omission of previous works, the following questions need answering:

- How does the VLM perform in terms of mAP? (Table 2)
- How does the Classifier perform without stage 1?
- What are the effects in Table 7?
- How does the method differ from previous VLMs/VLLMs in in-context emotion recognition, and how does it perform in comparison?

---

### Official Review · Reviewer_eUPn · 2025-10-30

**Soundness:** 3
**Presentation:** 3
**Contribution:** 3
**Rating:** 6
**Confidence:** 5

**Summary:**

The paper proposes a two-stage framework for context-aware emotion recognition from single images, leveraging instruction-tuned Vision-Language Models (VLMs). In Stage 1, an InstructBLIP-based VLM is adapted via a synthetically generated QA dataset to extract emotion-relevant visual features from three views: scene, body, and face, using shared but view-specific query tokens in the Q-Former. In Stage 2, the VLM is frozen, and its extracted features are fused via a lightweight self-attention classifier to produce calibrated, independent emotion scores. The method achieves state-of-the-art results on EMOTIC, CAER-S, and HECO without using engineered features like pose landmarks or human-object interaction.

**Strengths:**

1. Only the Q-Former and projection heads are tuned; the image encoder and LLM remain frozen.
2. Explicit disentanglement of scene/body/face contributions enables per-view analysis.
3. Unlike autoregressive VLMs, the final classifier provides independent confidence scores per emotion.
4. SOTA performance using only three basic visual inputs, outperforming methods with complex feature engineering.
5. Mitigates long-tail issues via minor-label QA augmentation and major-label downsampling.

**Weaknesses:**

1. All experiments use image-based datasets (EMOTIC, CAER-S, HECO); no video or temporal dynamics are considered, despite emotion being inherently dynamic in many real-world scenarios.
2. The related work and baselines stop at 2024, omitting key CVPR/ICML 2025 publications on VLMs and emotion recognition that would have been available by submission (Sept 2025).
3. The instruction-tuning corpus is fully auto-generated (via Osprey, Qwen-VL, Gemini); no manual verification of quality or bias is reported.
4. Critical hyperparameters (learning rate, batch size, optimizer, number of epochs) are missing, undermining reproducibility.
5. Despite emphasizing a "parameter-efficient" and "lightweight" design, the paper does not discuss computational cost or compare it to the state-of-the-art using these criteria.

**Questions:**

1. Given that CVPR/ICML/ACL 2025 proceedings were available by your submission date (Sept 2025), why were recent VLM-based emotion recognition methods not included in comparisons?
2. Can you provide quantitative metrics on model size, inference speed (ms/image), and GPU memory usage? How does your method compare to prior work in terms of FLOPs or real-time feasibility?
3. Was any subset of the synthetic QA dataset manually reviewed for emotional validity or factual consistency?
4. Please specify full training hyperparameters for both stages: optimizer, learning rate, batch size, number of epochs, and hardware used.

---

### Official Review · Reviewer_CN5h · 2025-10-31

**Soundness:** 3
**Presentation:** 3
**Contribution:** 2
**Rating:** 4
**Confidence:** 3

**Summary:**

1. The paper proposes a two-stage framework for context-aware emotion recognition from a single image: Stage-1 instruction-tunes a VLM into a multi-view emotion encoder (scene/body/face) with a frozen image encoder and LLM; Stage-2 freezes the VLM and trains a lightweight classifier with self-attention to fuse the three view descriptors and output calibrated per-label probabilities.

2. It introduces a label-aware data balancing scheme (minor-label QA augmentation + major-label down-sampling) and a synthetic instruction-tuning QA corpus.

3. The two step approach achieves good performance on several emotion recognition datasets.

**Strengths:**

- By freezing the image encoder and LLM while sharing a Q-Former, the approach is parameter-efficient.

- Competitive results using minimalist inputs (scene/body/face), reducing reliance on engineered signals.

**Weaknesses:**

In general, two-stage methods are not very convenient, especially that a large model is trained in stage 1.

And it is not clear how much the first stage is helping the second stage for the task of consideration. If the first stage is rephrasing the classification task with natural language, and the same vision encoder backbone is used in both stages, intuitively what additional information is gained by the classifier? Given that the VLM model actually works worse than the stage 2 classifier, it appears that aligning the modalities with stage 1 is hard.

In Table 2, is there a baseline where the classifier architecture remains the same, but without the stage 1 training? Such a baseline can tell how useful the first stage training is. I imagine the ``Ours (Classifier)'' refers to the model trained with both stages.

**Questions:**

See weakness.

---

### Note · Authors · 2025-12-29

I have read and agree with the venue's withdrawal policy on behalf of myself and my co-authors.